# Investigation on the Compressive Strength and Time of Setting of Low-Calcium Fly Ash Geopolymer Paste Using Response Surface Methodology

**DOI:** 10.3390/polym13203461

**Published:** 2021-10-09

**Authors:** Pauline Rose J. Quiatchon, Ithan Jessemar Rebato Dollente, Anabel Balderama Abulencia, Roneh Glenn De Guzman Libre, Ma. Beatrice Diño Villoria, Ernesto J. Guades, Michael Angelo Baliwag Promentilla, Jason Maximino C. Ongpeng

**Affiliations:** 1Center for Engineering and Sustainable Development Research, De La Salle University, Manila 1004, Philippines; ithan_dollente@dlsu.edu.ph (I.J.R.D.); anabel.abulencia@dlsu.edu.ph (A.B.A.); beatrice.villoria@dlsu.edu.ph (M.B.D.V.); michael.promentilla@dlsu.edu.ph (M.A.B.P.); 2Department of Chemical Engineering, De La Salle University, Manila 0922, Philippines; 3Department of Civil Engineering, De La Salle University, Manila 0922, Philippines; roneh_librejr@dlsu.edu.ph; 4Department of Civil Engineering, Technical University of Denmark, 2800 Kongens Lyngby, Denmark; ergua@byg.dtu.dk

**Keywords:** geopolymer paste, compressive strength, initial setting time, final setting time, Class F fly ash, RSM

## Abstract

Approximately 2.78 Mt of coal fly ash is produced in the Philippines, with a low utilization rate. Using fly ash-based geopolymer for construction will lessen the load sent to landfills and will result in lower GHG emissions compared to OPC. It is necessary to characterize the fly ash and optimize the geopolymer components to determine if it can replace OPC for in situ applications. The activator-to-precursor ratio, the water-to-solids ratio, and the sodium hydroxide-to-sodium silicate ratio were optimized using a randomized I-optimal design from the experimental results of 21 runs with five replicates, for a total of 105 specimens of 50 mm × 50 mm × 50 mm paste cubes. The engineering properties chosen as the optimization responses were the unconfined compressive strength (UCS), the initial setting time, and the final setting time. The samples were also ambient-cured with the outdoor temperature ranging from 30 °C to 35 °C and relative humidity of 50% ± 10% to simulate the on-site environment. Runs with high unconfined compressive strength (UCS) and short setting times were observed to have a low water-to-solids (W/S) ratio. All runs with a UCS greater than 20 MPa had a W/S ratio of 0.2, and the runs with the lowest UCS had a W/S of 0.4. The initial setting time for design mixes with a W/S ratio of 0.2 ranged from 8 to 105 min. Meanwhile, five out of seven design mixes with a W/S ratio of 0.4 took longer than 1440 min to set. Specimens with an alkali activator ratio (NaOH/WG) of 0.5 (1:2) and 0.4 (1:2.5) also had significantly lower setting times than those with an alkali activator ratio of 1. The RSM model was verified through confirmatory tests. The results of the confirmatory tests are agreeable, with deviations from the expected UCS ranging from 0 to 38.12%. The generated model is a reliable reference to estimate the UCS and setting time of low-calcium FA geopolymer paste for in situ applications.

## 1. Introduction

As of 2018, 55% of the world’s population lives in urban areas. By 2050, this statistic is projected to reach 68% [1]. As a consequence, more infrastructure is required to accommodate the additional people living in cities. Among the distinctive features of urban areas is their massive infrastructure, largely constructed using concrete. Concrete is globally accepted for its mechanical properties and cost-effectiveness compared to other construction materials. Furthermore, as it has been extensively researched, and we can control its mechanical and chemical properties for specific applications. However, it has been scrutinized for its contribution to global warming because of its primary component, cement, the production of which has been responsible for 7% of worldwide greenhouse gas emissions and 7% of industrial energy use [2]. The majority of greenhouse gases (GHG) produced are from clinker production, an essential process in cement production. Global cement production is forecasted to increase by 12–23% by 2050 [2].

Researchers have been looking for more environmentally friendly materials to replace OPC in concrete production. Among these are alkali-activated aluminosilicates or geopolymers [3]. The main components of geopolymers are the precursors, cement-like materials rich in silicon (Si), aluminum (Al), or iron (Fe) in an amorphous phase [4], and activators (alkaline solution). Although there are precursors available in nature, such as clays and kaolinite, it is highly recommended to source the precursors from waste byproducts (fly ash, silica fume, slag, rice hull ash) [4]. On the other hand, the activator can be composed of sodium hydroxide (NaOH), potassium hydroxide (KOH), sodium silicate (Na_2_SiO_3_), potassium silicate (K_2_SiO_3_), or a combination of these, which are all commercially available.

Approximately 52% of the power generated in the Philippines is from coal power plants [5]. As of 2020, there were approximately 52 coal power plants in the country—25 in Luzon, 14 in the Visayas, and 13 in Mindanao [6]. Presently, the coal fly ash (CFA) generated in the country amounts to ~2.78 Mt and is projected to reach ~13.02 Mt by 2035 [5]. With a low percentage of utilization of these CFA [7], the unused amount will be discarded and redirected to landfills. Though CFA is already commercially available and currently utilized for other applications (for example, earth stabilization) [8], it is preferable to penetrate the construction industry, as they are the largest consumer of OPC and, therefore, can have the largest impact with our push for sustainable development [9].

Transportation emissions are also a significant contributor to the total GHGs produced in the manufacturing of geopolymers [10]. Jamora et al. [5] recommended that the distance between the source (power plants) and the batching plant be less than 2841 km for the entire process to result in a net annual decrease in GHG emissions.

Unlike OPC, CFAs sourced from different coal power plants have different chemical compositions from one another, which will result in geopolymer concrete with varying engineering properties. Moreover, ASTM C618-19 [11] groups fly ash into two distinct categories: Class C (high calcium) fly ash is that with the calcium content comprising more than 10% of its mass, and Class F (low calcium) fly ash is that with the calcium content comprising less than 10%. Hence, it is essential to characterize these CFAs, especially in the Philippine setting, to determine their effect on the engineering properties of the resulting geopolymer concrete. Aside from the type of precursor, the mechanical properties of the geopolymer are also dependent on the type of alkaline activator, the proportioning of the ingredients, and the procedure for synthesizing the geopolymer.

The compressive strength and the setting time of the geopolymer paste are two essential characteristics that should be identified first when assessing the FA character in geopolymers, especially for in situ applications [12]. Using the response surface methodology, the factors affecting the compressive strength, the initial setting time, and the final setting time of fly ash-based geopolymers (with FA sourced from a Calaca coal power plant) can be further investigated and optimized. Section 1.1. elaborates on the existing research on the setting time of fly ash-based geopolymer and the parameters considered. Then, Section 1.2 summarizes the effects of adjusting the components of the geopolymer mixture on the compressive strength. Section 2 and Section 3 present the methodology and a discussion of the results, respectively. Finally, the conclusion and recommendations are discussed in Section 4.

### 1.1. Setting Time

Determining the initial and final setting time of both cement and geopolymer is necessary for in situ applications. The length of time from the moment where the cement makes contact with water to the time where it turns into a paste and begins to lose its plasticity is referred to as its initial setting time [13]. It should not be so fast that it becomes unworkable and difficult to cast; the standard minimum initial setting time for cement is 45 min [14]. On the other hand, the final setting time for cement is from the moment where it is mixed with water until it starts to gain structural strength and lose its plasticity [11]; this should not exceed 6.5 h so as to not slow down the construction process [14].

A study showed that the calcium content affects the setting time of fly ash-based geopolymers [15]. Low-calcium fly ash is known to have long setting times, especially without elevated-temperature curing to accelerate the polymerization process [15]. Hardjito et al. [16] reported that geopolymers cured at temperatures of 65 °C to 80 °C had initial and final setting times ranging from 129 min to 270 min. However, the samples left at room temperature needed at least a day to harden. Similarly, Elyamany et al. [15] observed that Class F fly ash geopolymer paste took 21–25 h to develop structural strength. Moreover, the addition of calcium-rich components, such as GGBS, significantly decreased the final setting time from one day to 100 min–150 min. Wijaya et al. [12] found that there was an exponential decrease in the setting time with an increase calcium content.

Other studies have also investigated the effect of varying NaOH concentrations on the time of setting. Mallikarjuna Rao and Gunneswara Rao [17] found that increasing the molarity of NaOH Solution from 8 M to 16 M of the low-calcium FA-based GP increased the setting time from 200 to 330 min. The NaOH content was the same for all mixes. On the other hand, the opposite trend was observed by Elyamany et al. [15]. The initial setting time for low-calcium FA geopolymer paste decreased from 180 min to 120 min as the NaOH molarity increased from 10 M to 16 M. Increasing the NaOH molarity increases the rate of geopolymerization by increasing the dissolution rate of Al^3+^ and Si^4+^ in fly ash. Ahmad Sofri et al. [18] also found the same trend for high-calcium FA but posited that the shorter setting time was due to the abundance of calcium silicate hydrate gel (CSH) from the reaction of the calcium content in the precursor and the silicate.

### 1.2. Compressive Strength

The existing literature on FA-based geopolymers reports that heat curing results in geopolymers with higher values of compressive strength because it accelerates the geopolymerization process [19]. Additionally, Garcia-Lodeiro et al. [20] posited that low-calcium precursors require harsh environments (for example, high alkalinity or high temperature) to be activated. However, this is neither feasible nor practical for in situ applications. Table 1 shows the studies on low-calcium FA and how it affects compressive strength. Vijai et al. [21] observed that the ambient-cured geopolymers took 28 days to obtain compressive strength close to those of heat-cured samples. Reed et al. [22] also reported a similar case—heat-cured geopolymer samples had better early-age mechanical properties (specifically, the samples had higher compressive strength). However, the 28-day compressive strength of ambient-cured geopolymers exceeded those of heat-cured samples. The polymerization for ambient-cured samples took longer compared to the heat-cured samples. The challenge for in situ applications is to properly characterize the fly ash and optimize the formulation to achieve the target compressive strength and setting time without relying on heat curing.

### 1.3. Response Surface Methodology

Ordinary Portland cement (OPC), one of the most well-known types of cement, strictly follows existing standards [27] that delimit its physical and chemical properties. The standardized chemical composition results in concrete mixes with fairly predictable engineering properties. On the other hand, the final engineering properties of alkali-activated aluminosilicates are dependent on the interaction of various factors, such as the chemical composition of the precursor, the amount of alkaline liquid and its component, and water content, among others. When investigating the interactions of these components, the use of traditional optimization techniques, such as one factor at a time (OFAT), by which only one factor will be kept constant to determine its effect on the product, is impractical. The response surface methodology combines mathematical and statistical techniques to produce polynomial equations that empirically explain the behavior of the data obtained from the experiment [28]. 

Existing studies on the optimization of cementitious composites have used classic response surface designs, such as central composite design (CCD) [29,30] and Box–Behnken design (BBD) [29,31]. Aside from these classic response surface designs, computer-aided designs, such as the I-optimal and D-optimal designs, are also increasing in popularity. CCD and BBD designs generate quadratic models based on the interactions of the factors. However, I-optimal and D-optimal designs do not limit the user to quadratic models and can also produce linear and cubic models [28]. Jones and Goos [32] compared the performance of I-optimal and D-optimal designs in locating the optimal point in the design region and found that the I-optimal model performed better. Longos et al. [33] conducted a similar investigation on the optimization of an FA-based geopolymer using an I-optimal design and produced an agreeable model, with deviations attributed to uncontrollable factors.

## 2. Materials and Methods

### 2.1. Materials

Fly ash was obtained from Pozzolanic Philippines Inc. (PPI), which sources its products from coal-fired power plants in Calaca, Batangas, Philippines. The fly ash was classified as low in calcium (Class F) following the test specifications of ASTM C618-19 [11]. The chemical and physical properties of the fly ash with the corresponding test specifications used are provided in Table 2 and Table 3, respectively. The properties were all provided by the manufacturer.

Sodium hydroxide flakes with 98% purity (manufactured by Formosa Plastic Corporation, Kaohsiung, Taiwan) and sodium silicate (waterglass) were combined in various proportions as the alkali activator. Table 4 shows the chemical composition of the analytical grade sodium silicate.

### 2.2. Parameters

A randomized I-optimal design was facilitated, and three factors—(a) the activator-to-precursor ratio, (b) the water-to-solids ratio, and (c) the NaOH-to-WG ratio—were used for the response surface study. Table 5 shows the factors used in the mixture design. A total of 21 runs were generated, with 5 replicates (Table 6).

#### 2.2.1. Activator-to-Precursor Ratio

Vora and Dave [23] reported that the variation of the A/P ratio had no significant effect on the compressive strength. Elyamany et al. [15], on the other hand, found that the compressive strength was inversely related to the A/P ratio. However, multiple researchers have observed that the compressive strength increased with the A/P ratio, peaked at an A/P of 0.4, and subsequently decreased [16,34]. The authors of the present study used these existing reports [15,16,23,24,34,35] as a basis for the A/P values, which varied from 0.3 to 0.5. The activator used was an alkaline solution of NaOH and sodium silicate (Waterglass).

#### 2.2.2. Water-to-Rolids Ratio

Previous studies have demonstrated that there is a clear inverse relationship between the W/S ratio and the compressive strength, with the W/S ratio ranging from 0.17 to 0.34 [16,19,23,34]. Water-to-solids ratios of 0.2 and 0.3 were chosen for this experiment based on these findings. A W/S ratio of 0.4 was also included to check if the resulting compressive strength is acceptable in other applications as a more economical option.

#### 2.2.3. NaOH-to-Waterglass Ratio

Longos et al. [33] reported that among that the three NaOH-to-WG ratios in their experiment (0.5, 1, 2), a NaOH/WG ratio of 0.5 resulted in the highest compressive strength. A low NaOH/WG ratio means that there was a high amount of SS that supplied more SiO_3_ in the mixture, which is a valuable component for the geopolymerization reaction. However, lowering the NaOH/WG ratio to 0.4 resulted in lower values of compressive strength [23]. For this experiment’s design mix, NaOH/WG ratios of 0.4, 0.5, and 1.0 were used. Even though it was expected that a NaOH/WG ratio of 1 would result in lower values of compressive strength, it is the less costly option among the three.

### 2.3. Experimental Procedure/Geopolymer Synthesis

From the factors in Table 6, the mass of each component was calculated following the flowchart in Figure 1. To start with, the mass of the fly ash needed for a specified volume was estimated—in this case, the authors used 1300 g of fly ash to fill five 50 mm × 50 mm × 50 mm molds and one conical ring. Then, the masses of the other components (such as the NaOH flakes, waterglass, and water) were computed following the steps in Figure 1.

For the preparation of the geopolymer, the fly ash and NaOH flakes were weighed beforehand. The NaOH flakes were then dissolved in tap water and placed over an ice bath for 15 min or until the temperature of the solution reached 27 °C. Once the NaOH solution was cooled down, the waterglass was subsequently weighed and added to the solution, continuously stirring it for five minutes. The alkaline solution (NaOH and waterglass) and the fly ash were then mixed for 8–10 min using a JJ-5 cement mortar mixer. An extra 1 min of manual mixing followed to ensure homogeneity and that no clumps of dry ingredients remained in the mixture. After mixing, the paste was first poured into the conical ring for the setting time and then into 2 sets of 50 mm × 50 mm × 50 mm polyethylene square molds. Each mold has three cube compartments. The cubes were compacted by tamping two layers to ensure that there were no air bubbles within the paste. For quality control, a putty knife was used to flatten the top of the paste. The specimen was allowed to rest for 24 h before demolding. The demolded specimens were left in an undisturbed area, protected from rain and direct sunlight. The ambient temperature was noted to vary between 30 °C and 35 °C, with a relative humidity of 50 ± 10%. The researchers applied the existing guidelines for OPC in this study. The ASTM C109/C109M method [36] was used for casting, quality control, and unconfined compressive strength (UCS) testing. The ASTM C191 method [13] was used for the determination of initial and final setting times.

### 2.4. Unconfined Compressive Strength (UCS)

The 28-day compressive strength of concrete is among the primary mechanical properties considered when assessing the performance of concrete. The unconfined compressive strength (UCS) test was conducted following the standards outlined in ASTM C109/C109M [36] using MATEST SpA Treviolo (250 KN) with a loading rate of 0.9 KN/s. ASTM C109/C109M covers the determination of the compressive strength of a 2-in [50-mm] paste cube. Similar to OPC-based paste/mortar, compressive strength is the material’s capacity to carry the applied load before failure [37]. The unconfined compressive strength of each individual sample is equal to the maximum load recorded by the testing machine divided by the area of the surface of the sample in contact with the machine (Equation (1)) [36].
(1)$$\mathrm{Unconfined}\;\mathrm{Compressive}\;\mathrm{strength}\;\left(\mathrm{MPa}\right)=\frac{\mathrm{Maximum}\;\mathrm{load}\;\left(N\right)}{\mathrm{Loaded}\;\mathrm{Area}\;({\mathrm{mm}}^{2})\;}$$

### 2.5. Initial and Final Setting Time

The initial setting time was determined as per ASTM C191 [13]. This test method covers the setting of hydraulic cement by means of a Vicat needle. Periodic penetration tests were performed on the geopolymer paste by allowing a 1-mm Vicat needle to settle into the paste. The Vicat initial time setting was calculated as the time elapsed between the initial contact of fly ash and alkaline solution and the time when penetration was at 25 mm. The Vicat final time of setting was calculated as the time elapsed between initial contact of fly ash and alkaline solution and the time when the needle could not penetrate the paste any further.

## 3. Results and Discussion

The factors and responses for the 21-run design of the experiment are shown in Table 7. The response parameters are the initial setting time, the final setting time, and compressive strength. Five out of 21 runs had compressive strength values above 20 MPa, and 10 out of 21 had initial and final setting times within the recommended durations [14]. The initial setting time was arranged from the lowest value to the largest, and a boxplot of the compressive strength values was overlaid to demonstrate the trend between the two parameters. It can be observed that runs with low unconfined compressive strength values took longer to set (Figure 2). On the other hand, runs with low water-to-solids ratios were faster to set and had high values of compressive strength. The common factor of the response parameters is the water-to-solids ratio, which is discussed further in Section 3.1 and Section 3.2.

### 3.1. Factors Affecting Initial and Final Setting Times

The most significant factor affecting the initial and final setting times is the water-to-solids ratio, as it has the highest correlation. The correlation coefficients for the initial and final setting times with the W/S ratio are 0.7571 and 0.7371, respectively. It can be observed that as the water-to-solid ratio increased, the setting time increased accordingly. This trend corresponds with the results obtained by Vora and Dave [23] and Rangan [26]. The increased water contributed to a less viscous consistency and contributed to more dissolution of the activator solution.

Runs with a W/S ratio of 0.2 had initial setting times between 8 min to 105 min. These were considerably low compared to other findings on low-calcium FA-based geopolymers that were ambient cured with W/S ratios of 0.2 [34]. On the high end of the spectrum, runs with W/S ratios of 0.4 took the longest to set. Five out of seven runs had initial setting times longer than one day.

The curve along the axis for the water-to-solids ratio has the steepest slope (Figure 3a–c). That is, the variation in the water-to-solids ratio had the most significant effect on the initial setting time. Changing the NaOH-to-WG ratio and activator-to-precursor ratio had a minimal effect on the duration of the initial setting time if the W/S ratio was 0.2 (Figure 3a–c). On the other hand, for high W/S ratios, varying the NaOH-to-WG ratio or the A/P ratio had more observable effects on the initial setting times. Additionally, a NaOH-to-WG ratio of 1 (Figure 3c) yielded a higher initial setting time compared to those with NaOH-to-WG ratios of 0.4 and 0.5 (Figure 3a,b). The response surface graphs of the final setting time follow that of the initial setting time, as they were very closely correlated (R^2^ = 0.958).

The authors also observed that if the paste with a W/S ratio of 0.2 was not immediately cast into the molds after mixing, the resulting specimens exhibited honeycombing. Honeycombing occurs in cast-in-place concrete when the mortar cannot flow between the coarse aggregates and fills the space below. Thus, the surface of the hardened concrete will have voids in those spaces not filled with mortar [38]. Superplasticizers can be incorporated in future mixes to improve the workability of the mix due to the better dispersion of the components [23]. Hardjito et al. [19] and Vora and Dave [23] have suggested using a superplasticizer of approximately 2% of fly ash by mass. The dosage can increase up to 4%, but minor reductions in the compressive strength were reported beyond 2% [19,23].

### 3.2. Factors Affecting Compressive Strength

In the same way as with the initial and final setting times, the water-to-solids ratio of the mixture had a strong correlation with the compressive strength (Figure 4a–c). Hardjito et al. [19], Vora and Dave [23], and Barbosa et al. [39] observed similar trends in their investigations. The increase in water leads to a decrease in the bulk density of the mixture. Additionally, OPC concrete exhibits a similar response, where the water-to-cement ratio significantly affects its compressive strength; although, the interaction between the components and the effects of such interactions for geopolymer and OPC are somewhat different [19]. According to Zhang et al. [40], the compressive strength changes with any variation in the reaction product (gel), residual particles, and the pore structure. However, the pore structure is significantly related to the mechanical properties, especially at an early age [20]. The water-to-solids ratio significantly affects the porosity of the geopolymer matrix at the interfacial transition zones (ITZs), as the increase of water creates vulnerable points, leading to lower strengths.

The unconfined compressive strength values of the geopolymer specimens were observed to range from 1.07 MPa to 29.412 MPa. The values presented in Table 7 are the mean compressive strength of each mix design, with the outliers removed using the procedure outlined in ASTM C109 [36]. Runs 2 and 11 did not have values of compressive strength because all of the samples split before testing.

A water-to-solids ratio of 0.4 only gave compressive strength values of less than 10 MPa, even if the A/P ratio varied from 0.3 to 0.5 and the NaOH/WG ratio varied from 0.4 to 1.0. Five out of seven mixtures with W/S ratios of 0.2 had mean compressive strength values higher than 20 MPa.

### 3.3. Response Surface Analysis

The statistical analysis of the initial and final setting times followed a natural logarithmic transform (y’ = ln(y + k)), where the constant k was equal to 0 for the correlation. This was based on the recommended standard transformation by Design Expert 11 (Design-Expert^®^ software, version 11), using the current data to achieve at least a 95% confidence interval for the model. The regression models are shown as follows:*ln (Initial Setting Time)* = 4.995 + 0.126*A* + 1.746*B* + 0.058*C* − 1.274*AB* + 0.714*AC* + 0.782*BC*(2)
*ln (Final Setting Time)* = 5.565 + 0.189*A* + 1.595*B* + 0.187*C* − 1.119*AB* + 0.660*AC* + 0.600*BC*(3)
*ln (Compressive Strength)* = 2.281 + 0.095*A* − 0.675*B* + 0.145*C*(4)
where A = activator-to-precursor ratio; B = water-to-solids ratio; C = NaOH-to-WG ratio.

The P-value of the initial setting time is slightly lower than the final setting time, which means that the overall model is significant (Table 8). The model F-values for the equations of the initial setting time, the final setting time, and compressive strength are statistically significant, meaning that there is little chance that the model values would occur due to noise. Moreover, the P-values are all less than 0.05, which means that the combination of the three regression models are statistically significant. Lastly, the lack-of-fit values for the three factors are all greater than 0.05, indicating that lack of fit is insignificant and implying a good model fit. These observations are important to assess if the model can still be used to navigate the design space, even if the R-squared values are only about 0.7 to 0.76.

The details of the analysis of variance (ANOVA) for the three response parameters can be found in Table 9, Table 10 and Table 11.

### 3.4. Qualitative Assessment of Geopolymer Samples

White, powdery deposits were observed on some of the specimens. This phenomena, referred to as efflorescence, occur because of several mechanisms [41], including soluble constituents present in the specimen, water passing through the specimen (to dissolve the soluble constituents), and a force that allows the movement of water within the specimen (gravity, capillary action, hydrostatic pressure, or evaporation).

For OPC-based concrete, efflorescence is merely an aesthetic issue, as can be seen in Figure 5a. It is usually controlled by limiting the alkalinity of the water used in the mix formulation [42]. However, controlling the formation of efflorescence may be more challenging for geopolymers since geopolymers generally have higher alkaline content than cement-based concrete, and an alkaline solution is required for the polymerization process to take place [3,4]. Efflorescence can also form beneath the surface of the material and cause defects that can affect the structural integrity of the material [25,41,43]. Subflorescence, an extended form of efflorescence, happens when salt precipitates within the geopolymer matrix [25,43]. Since the crystal growths are confined within the matrix, they can cause pressure on the surrounding material and may lead to cracking if the stress is high enough to overcome the tensile strength of the material [25].

As discussed earlier, 2 out of 21 samples were not tested for compressive strength because the specimens cracked before the day of testing (Figure 5b). Visual inspection of the post-cracked samples showed that there were existing salt deposits on the plane of failure of the specimen (Figure 6a,b). This implies that subflorescence took place in the sample.

As of now, little research has been done on efflorescence control that does not involve heat curing.

### 3.5. Confirmatory Run

An optimized mix formulation was obtained by setting the initial and final setting time limits and maximizing the target compressive strength. Three calculated optimized runs were synthesized and performed as in the procedure in Section 2.3 (Table 12). The observed compressive strength values for optimization runs 2 and 3 had significantly higher values than the model prediction. Nonetheless, both values are well within the 95% confidence interval of the optimization’s upper limits, and they validate the model’s predictability. The positive deviation may have been caused by uncontrolled and immeasurable human factors, such as the variation in persons who performed the confirmatory runs and changes in ambient curing conditions. Run 2 had the best response result of 23.08 MPa, based on the maximum compressive strength.

### 3.6. Possible In Situ Applications

The standard initial setting time for practical applications of paste should be more than 45 min, and the final setting time should be less than 6.5 h (390 min) [14]. Ten out of 21 mixtures have setting times within the suggested range, five of which have a water-to-solids ratio of 0.2. Some of the possible in situ applications for OPC mortar and concrete with compressive strength comparable to the resulting compressive strength of the geopolymer are summarized in Table 13. Only these mixtures are considered for in situ applications. Mixes with low compressive strength (10–15 MPa) can be used to repair masonry in historical structures as well. The compressive strength values of the materials in masonry structures are lower than that of modern OPC-concrete buildings. The standard also requires that the compressive strength of the repair material is lower than the existing masonry [42]. Additionally, from the optimized runs for in situ application (Table 12), it can be observed that the initial and final setting times were all within the recommended durations [14].

The development of fly ash-based geopolymer as a replacement for OPC is essential in progressing the use of sustainable construction materials in the Philippines and alleviating some existing problems (such as fly ash waste management). The existing research can motivate construction companies, as they are the largest stakeholders, to consider using alternatives to OPC and provide guidance on the use and manufacturing of geopolymers [9]. An example of these studies is the one reported by Ongpeng et al. [47], in which they investigated the feasibility of using CFA-based geopolymer concrete as a structural member.

## 4. Conclusions and Recommendations

Geopolymer is an attractive replacement for OPC because its manufacturing emits lower GHGs compared to OPC. The engineering properties of geopolymer mortar and concrete are comparable to traditional OPC-based products. Recent studies recommend elevated temperature curing for the development of high early-age compressive strength. However, this is impractical for in situ applications or cast-in-place concrete. A randomized I-optimal design was applied, and three factors (activator-to-precursor ratio, water-to-solids ratio, and sodium hydroxide-to-waterglass ratio) were varied to determine their effects on the response parameters (compressive strength, initial and final setting times) of the geopolymer paste. The water-to-solids ratio was the primary factor, with the most significant effect on all three response parameters. A lower W/S ratio (0.2) resulted in higher compressive strength values and faster setting time. Adversely, a high W/S ratio (0.4) led to lower compressive strength values and longer setting times. Five out of 21 runs achieved an average 28-day compressive strength value of 20 MPa and above. Ten out of 21 runs had initial setting times above 45 min and final setting times below 6.5 h. Though design mixes with a W/S ratio of 0.2 had excellent engineering properties, comparable to OPC, the paste was unworkable, and honeycombing was observed if the paste was not immediately cast into the molds.

A confirmatory experiment was performed to verify the RSM Model. Both the initial and final setting times of all the runs in the confirmatory test were within the recommended durations [14]. The results for the unconfined compressive strength (UCS) tests were also agreeable, with deviations of up to +38.12%.

To enhance the workability of the mix for in situ applications, it is recommended to consider using superplasticizers, as long as any adverse effects on the engineering properties are examined beforehand. Mass loss from drying can lead to shrinkage, which, in turn, may cause hairline cracks [48]. This property should be analyzed closely for future research on in situ applications. The addition of slag or other calcium-rich materials should also be explored to investigate the improvement in the resulting engineering properties.

## Figures and Tables

**Figure 1 polymers-13-03461-f001:**
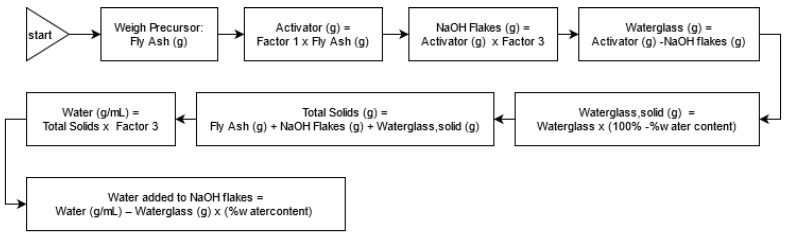
Guideline for the computation of the mass of each component for each run from the factors in Table 6.

**Figure 2 polymers-13-03461-f002:**
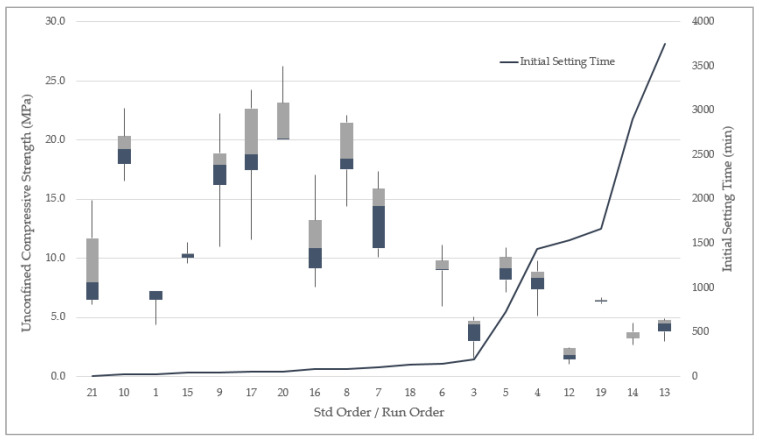
Unconfined compressive strength (boxplot) and initial setting time (line graph) per run.

**Figure 3 polymers-13-03461-f003:**
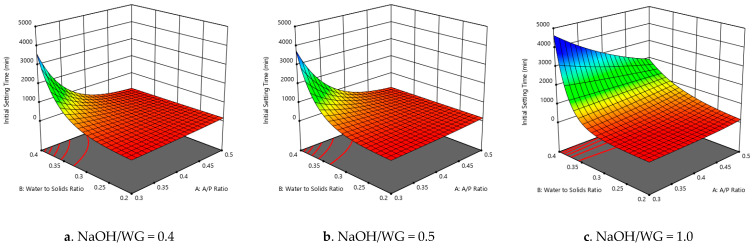
(**a**–**c**) Initial setting time vs. water-to-solids ratio and A/P ratio for each NaOH/WG ratio value.

**Figure 4 polymers-13-03461-f004:**
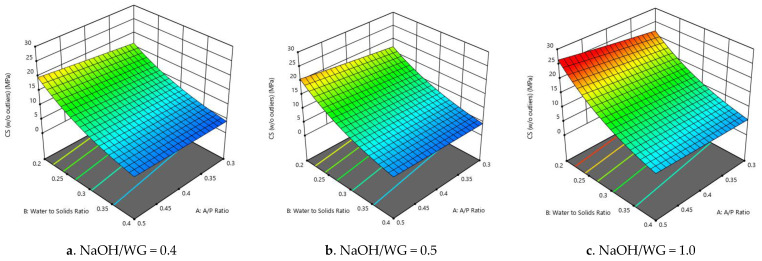
(**a**–**c**) Compressive strength vs. water-to-solids ratio and A/P ratio for each NaOH/WG ratio value.

**Figure 5 polymers-13-03461-f005:**
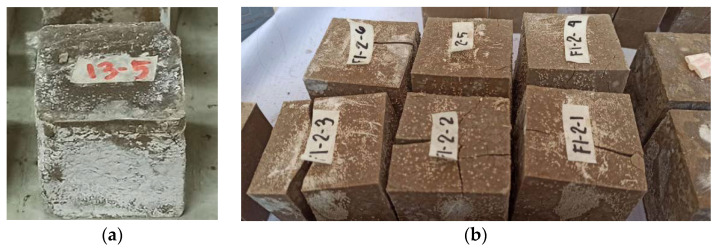
(**a**) Sample of a specimen that exhibits efflorescence. (**b**) Specimens that exhibit efflorescence and cracked before testing.

**Figure 6 polymers-13-03461-f006:**
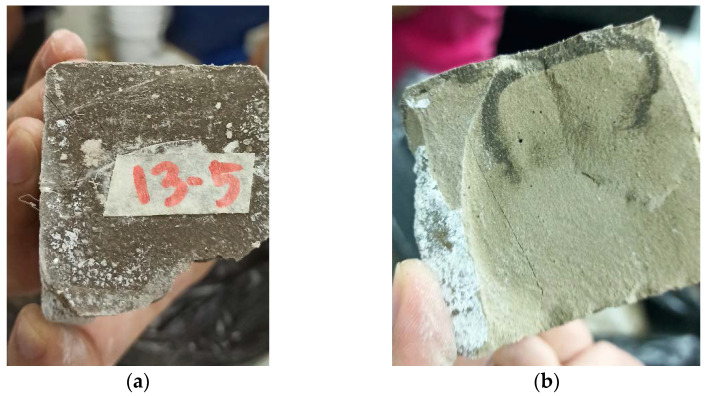
Evidence of subflorescence on sample 5 of mix 13. (**a**) Top View of the sample (post-test) (**b**) Plane of failure of the sample after compressive test, showing existing crystal deposit within the matrix.

**Table 1 polymers-13-03461-t001:** Studies on low-calcium FA geopolymers and the factors that affect the compressive strength.

Factor	Explanation	Reference
Concentration of NaOH (in terms of molarity	Direct relationship: an increase in molarity leads to an increase in compressive strength (ideal range: 8 M to 16 M)	[15,17,19,23]
Sodium silicate-to-NaOH ratio (by mass)	Direct relationship: higher SS/SH ratio provides higher compressive strength (because of high waterglass content).	[19,23,24,25]
Curing temperature	Heat-cured geopolymer possessed higher early-age compressive strength than ambient-cured geopolymer. Optimal temperature = 60 °C.	[19,23]
Curing time (for heat curing)	Longer curing time results in higher compressive strength but curing beyond 48 h is no longer practical because of the minimal increase in strength.	[19,23]
Water-to-solids ratio	Inverse relationship. As the ratio of water-to-geopolymer solids by mass increases, the compressive strength of geopolymer mortar decreases.	[16,19,23]
Age	For ambient-cured samples, the age of geopolymer is essential, as it takes time for the geopolymerization to occur. For heat-cured samples, the target compressive strength is mostly attained within 7 days.	[22,26]
Alkaline liquid to fly ash	Several authors have reported different findings regarding this factor:	
	1. It has no significant effect;	[23]
	2. A higher A/F ratio leads to a decrease in compressive strength;	[15]
	3. A higher A/F ratio leads to an increase in compressive strength with an optimal value of A/F = 0.4.	[16,24]
Plasticizer	Commonly used: naphthalene-based superplasticizer. Dosage beyond 2% by mass reduces compressive strength.	[19,23]

**Table 2 polymers-13-03461-t002:** Chemical composition of fly ash (provided by manufacturer).

Chemical Properties	(%)	Test Methods
Silicon Dioxide (SiO_2_)	57.2	ASTM C114
Aluminum Trioxide (Al_2_O_3_)	21.8	
Ferric Oxide (Fe_2_O_3_)	4.73	
Calcium Oxide (CaO)	6.9	
Magnesium Oxide (MgO)	9.9	
Loss on Ignition (LOI)	0.6	ASTM C311
Insoluble Residue (IR)	55.1	ASTM C114
Sulfur Trioxide (SiO_3_)	1.23	
Moisture Content	0.1	

**Table 3 polymers-13-03461-t003:** Physical properties of fly ash (provided by manufacturer).

Physical Properties	Values		Test Methods
Fineness, Retained on 45 μm Sieve	12.8		ASTM C430/C311 (sec 20)
Autoclave Expansion (%)	0.07		ASTM C151/C311 (sec 24)
Autoclave Contraction (%)			
Density (g/cm^3^)	2.27		ASTM C188/C311 (sec 19)
Strength Activity Index: (%)			ASTM C311 (sec 27, 28, 29)
With Portland Cement, 7 days	83.8		
With Portland Cement, 28 days	97.6		
Control Mix			
Portland Cement, 7 days	33.0 MPa	4790 psi	
Portland Cement, 28 days	44.7 MPa	6490 psi	
Water Requirement (%)	95		ASTM C311 (sec 31)

**Table 4 polymers-13-03461-t004:** Chemical composition of waterglass.

Chemical Composition	Content
SiO_2_	34.13%
Na_2_O	14.65%
H_2_O	51.22%
Silica Modulus	2.33

**Table 5 polymers-13-03461-t005:** Parameters of each factor.

Factors	Low Level	Mid-Level	High Level
Factor 1: Activator-to-precursor ratio	0.3	interval	0.5
Factor 2: Water-to-solids ratio	0.2	0.3	0.4
Factor 3: NaOH-to-waterglass ratio	0.4	0.5	1

**Table 6 polymers-13-03461-t006:** Experiment design (21 runs).

Std Order/ Run Order	Factor 1: Activator-to-Precursor Ratio	Factor 2: Water-to-Solids Ratio	Factor 3: NaOH-to-WG Ratio
1	0.5	0.40	0.50
2	0.5	0.3	1.00
3	0.5	0.3	0.40
4	0.346	0.4	1.00
5	0.454	0.3	1.00
6	0.4	0.3	1.00
7	0.454	0.2	0.50
8	0.453	0.2	0.40
9	0.4	0.2	0.40
10	0.3	0.2	0.50
11	0.4	0.4	0.50
12	0.3	0.4	0.40
15	0.3	0.4	1.00
14	0.4	0.4	0.40
15	0.5	0.3	0.50
16	0.4	0.3	0.50
17	0.454	0.2	0.50
18	0.377	0.4	0.40
19	0.5	0.4	1.00
20	0.345	0.2	0.40
21	0.348	0.2	0.50

**Table 7 polymers-13-03461-t007:** Design data and results.

Std Order/ Run Order	Factor 1: (A/P)	Factor 2: (W/S)	Factor 3: (NaOH/WG)	Response 1: Initial Setting Time (min)	Response 2: Final Setting Time (min)	Response 3: Compressive Strength (MPa)	Setting Time Within Recommended Duration? [14]
1	0.50	0.4	0.5	30	85	7.20	✗
2	0.50	0.3	1	193	498	-	✗
** *3* **	** *0.50* **	** *0.3* **	** *0.4* **	** *195.09* **	** *312.8* **	** *5.33* **	** *✓* **
4	0.35	0.4	1	1441.86	2076.96	9.51	✗
5	0.45	0.3	1	723.38	1276	9.03	✗
** *6* **	** *0.40* **	** *0.3* **	** *1* **	** *146.29* **	** *339.31* **	** *9.20* **	** *✓* **
** *7* **	** *0.45* **	** *0.2* **	** *0.5* **	** *105.93* **	** *154.21* **	** *16.64* **	** *✓* **
** *8* **	** *0.45* **	** *0.2* **	** *0.4* **	** *86.25* **	** *209.96* **	** *21.78* **	** *✓* **
** *9* **	** *0.40* **	** *0.2* **	** *0.4* **	** *49* **	** *86* **	** *23.24* **	** *✓* **
*10*	*0.30*	*0.2*	*0.5*	*28.14*	*49*	*24.13*	**✗**
11	0.40	0.4	0.5	1375	1417	-	✗
12	0.30	0.4	0.4	1533	1786	2.44	✗
13	0.30	0.4	1	3747	4423	4.72	✗
14	0.40	0.4	0.4	2897	5730.33	3.43	✗
** *15* **	** *0.50* **	** *0.3* **	** *0.5* **	** *45* **	** *49* **	** *10.09* **	**✓**
** *16* **	** *0.40* **	** *0.3* **	** *0.5* **	** *80.66* **	** *129* **	** *9.23* **	**✓**
** *17* **	** *0.45* **	** *0.2* **	** *0.5* **	** *52* **	** *70* **	** *23.48* **	**✓**
** *18* **	** *0.38* **	** *0.4* **	** *0.4* **	** *133* **	** *238* **	** *6.00* **	**✓**
19	0.50	0.4	1	1671	2516	6.41	✗
** *20* **	** *0.35* **	** *0.2* **	** *0.4* **	** *56.55* **	** *88* **	** *20.20* **	** *✓* **
21	0.35	0.2	0.5	8.34	12	6.86	✗

**Table 8 polymers-13-03461-t008:** Summary of ANOVA results for the three regression models.

Equation	F-Value	*p*-Value	Mean	Std. Dev.	Lack of Fit	R-Squared
Equation (2)	7.27	0.0011	5.26	1.03	0.3576	0.76
Equation (3)	6.54	0.0019	5.75	1.04	0.3915	0.74
Equation (4)	11.41	0.0004	2.24	0.41	0.4281	0.70

**Table 9 polymers-13-03461-t009:** ANOVA for reduced 2FI model (Response 1: initial setting time).

Source	Sum of Squares	df	Mean Square	F-Value	*p*-Value	
Model	46.17	6	7.70	7.27	0.0011	significant
A-A/P Ratio	0.1169	1	0.1169	0.1105	0.7445	
B-Water to Solids Ratio	16.91	1	16.91	15.98	0.0013	
C-NaOH/WG Ratio	0.0239	1	0.0239	0.0226	0.8826	
AB	8.27	1	8.27	7.82	0.0143	
AC	2.61	1	2.61	2.47	0.1384	
BC	2.35	1	2.35	2.22	0.1585	
Residual	14.81	14	1.06			
Lack of Fit	14.56	13	1.12	4.42	0.3576	not significant
Pure Error	0.2531	1	0.2531			
Cor Total	60.99	20				

**Table 10 polymers-13-03461-t010:** ANOVA for reduced 2FI model (Response 2: final setting time).

Source	Sum of Squares	df	Mean Square	F-Value	*p*-Value	
Model	42.13	6	7.02	6.54	0.0019	significant
A-A/P Ratio	0.2622	1	0.2622	0.2442	0.6288	
B-Water to Solids Ratio	14.11	1	14.11	13.14	0.0028	
C-NaOH/WG Ratio	0.2452	1	0.2452	0.2284	0.6401	
AB	6.38	1	6.38	5.95	0.0287	
AC	2.23	1	2.23	2.08	0.1714	
BC	1.38	1	1.38	1.29	0.2759	
Residual	15.03	14	1.07			
Lack of Fit	14.72	13	1.13	3.63	0.3915	not significant
Pure Error	0.3119	1	0.3119			
Cor Total	57.16	20				

**Table 11 polymers-13-03461-t011:** ANOVA for linear model (Response 3: compressive strength).

Source	Sum of Squares	df	Mean Square	F-Value	*p*-Value	
Model	5.78	3	1.93	11.41	0.0004	significant
A-A/P Ratio	0.0814	1	0.0814	0.4814	0.4984	
B-Water to Solids Ratio	5.49	1	5.49	32.47	<0.0001	
C-NaOH/WG Ratio	0.2311	1	0.2311	1.37	0.2606	
Residual	2.54	15	0.1690			
Lack of Fit	2.48	14	0.1769	2.98	0.4281	not significant
Pure Error	0.0593	1	0.0593			
Cor Total	8.32	18				

**Table 12 polymers-13-03461-t012:** Optimized runs for in situ application.

Run	Factor 1: (A/P)	Factor 2: (W/S)	Factor 3: (NaOH/WG)	Response 1: Initial Setting Time (min)	Response 2: Final Setting Time (min)	Predicted UCS (MPa)	Response 3: Observed UCS (MPa)	Deviation
O1	0.381	0.222	0.401	60	130.0	16.75	16.75	0%
O2	0.329	0.223	0.559	47.5	87.4	16.71	23.08	38.12%
O3	0.361	0.228	0.422	57.5	106.8	16.35	21.51	31.56%

**Table 13 polymers-13-03461-t013:** Applications of OPC with compressive strength.

Material	Minimum UCS (MPa)	Application	Source
Class A Concrete	20.7	Concrete structures and concrete pavement	DPWH and ASTM Standards [44,45] as summarized in Longos et al. [33]
Class B Concrete	16.5	Pedestrian and light-traffic pavement	
Class C Concrete	20.7	Plain concrete for structures	
Class F Concrete	11.8	Plain concrete for leveling	
Class R1	10	Repair mortar	Ducman et al. [46]
Class R2	15	Repair mortar	

## Data Availability

Not applicable.

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
