# Peer review of "Investigation on the Compressive Strength and Time of Setting of Low-Calcium Fly Ash Geopolymer Paste Using Response Surface Methodology"

_polymers, 2021, doi:10.3390/polym13203461_

Round 1

Reviewer 1 Report

In this paper, the compressive strength and setting time of geopolymers are studied by the response surface methodology. There are a few questions:
1. The serial number of table. Parameters of each factor and table Experiment Design (21 runs) needs to be modified.
2. The introduction part is not detailed, and the advantages of response surface methodology should be explained more. The following references can be cited:
https://doi.org/10.1016/j.jobe.2020.102101
https://doi.org/10.1016/j.cscm.2021.e00691
https://doi.org/10.1016/j.matdes.2015.07.049
3. Table. Experiment Design. It should be specified that Factor 1 is an interval, not three fixed values.
4. Explain in detail the reasons for the selection of the three factors. For example, why factor 2 is 0.2, 0.3, and 0.4. Why factor 3 is 0.4, 0.5, and 1.
  5. When researching through response surface methodology, there should be a verification process. Please supplement the verification experiment and explain it. Instead of giving a possible mix ratio (Table 10).
6. In the part of the experimental results, the mechanism discussion and analysis of compressive strength and setting time should be detailed.

Reviewer 2 Report

The paper is very interesting, well written and well structured. The methodological rigor and the scientific profile are not lacking. Allow me to suggest some additions to improve the attractiveness of the paper to the scientific community.

*) Figure 1 is extremely interesting so it deserves a more detailed caption in order to be self-explanatory.

*) Please highlight the most relevant numerical values ​​shown in the Tables in bold.

*) How did the authors define Compressive Strength? Please specify.

*) Polymer Pastes are often used for the production of membranes for the most diverse uses including membranes for micro-electrostatic devices such as MEMS membranes. So, to highlight the great versatility of these materials, I recommend inserting a sentence in the text of the Introduction of the paper that highlights this versatility by putting the following relevant works in the bibliography:

doi: 10.3390/s21155237

doi:  10.1002/adem.201700858

Round 2

Reviewer 1 Report

The paper has been modified, and it can be published!